# High Thioredoxin Domain-Containing Protein 11 Expression Is Associated with Tumour Progression in Glioma

**DOI:** 10.3390/ijms241713367

**Published:** 2023-08-29

**Authors:** Ying-Tso Chen, Chia-Li Chung, Yu-Wen Cheng, Chien-Ju Lin, Tzu-Ting Tseng, Shu-Shong Hsu, Hung-Pei Tsai, Aij-Lie Kwan

**Affiliations:** 1Division of Neurosurgery, Department of Surgery, Kaohsiung Veterans General Hospital, Kaohsiung 813414, Taiwan; a7513366@yahoo.com.tw (Y.-T.C.); murraycheng1015@gmail.com (Y.-W.C.); haussoo67@gmail.com (S.-S.H.); 2Graduate Institute of Medicine, College of Medicine, Kaohsiung Medical University, Kaohsiung 807, Taiwan; 3Department of Surgery, Kaohsiung Municipal Siaogang Hospital, Kaohsiung 81267, Taiwan; r1chung@yahoo.com.tw; 4School of Pharmacy, College of Pharmacy, Kaohsiung Medical University, Kaohsiung 807, Taiwan; mistylin@kmu.edu.tw; 5Division of Neurosurgery, Department of Surgery, Kaohsiung Medical University Hospital, Kaohsiung 807, Taiwan; cawaii7992@gmail.com; 6Department of Surgery, School of Medicine, College of Medicine, Kaohsiung Medical University, Kaohsiung 807, Taiwan

**Keywords:** GBM, TXNDC11, prognosis, TMZ-sensitivity

## Abstract

Glioblastoma (GBM) is the most common primary brain malignancy in adults. Despite multimodal treatment that involves maximal safe resection, concurrent chemoradiotherapy, and tumour treatment for supratentorial lesions, the prognosis remains poor. The current median overall survival is only <2 years, and the 5-year survival is only 7.2%. Thioredoxin domain-containing protein 11 (TXNDC11), also known as EF-hand binding protein 1, was reported as an endoplasmic reticulum stress-induced protein. The present study aimed to elucidate the prognostic role of TXNDC11 in GBM. We evaluated the clinical parameters and TXNDC11 scores in gliomas from hospitals. Additionally, proliferation, invasion, migration assays, apoptosis, and temozolomide (TMZ)-sensitivity assays of GBM cells were conducted to evaluate the effects of short interfering RNA (siRNA) on these processes. In addition, these cells were subjected to Western blotting to detect the expression levels of N-cadherin, E-cadherin, and Cyclin D1. High levels of TXNDC11 protein expression were significantly associated with World Health Organization (WHO) high-grade tumour classification and poor prognosis. Multivariate analysis revealed that in addition to the WHO grade, TXNDC11 protein expression was also an independent prognostic factor of glioma. In addition, TXNDC11 silencing inhibited proliferation, migration, and invasion and led to apoptosis of GBM cells. However, over-expression of TXNDC11 enhanced proliferation, migration, and invasion. Further, TXNDC11 knockdown downregulated N-cadherin and cyclin D1 expression and upregulated E-cadherin expression in GBM cells. Knock-in TXNDC11 return these. Finally, in vivo, orthotopic xenotransplantation of TXNDC11-silenced GBM cells into nude rats promoted slower tumour growth and prolonged survival time. TXNDC11 is a potential oncogene in GBMs and may be an emerging therapeutic target.

## 1. Introduction

Glioblastoma (GBM) is a devastating and intractable type of brain cancer. It is the most common malignant central nervous system (CNS) tumour, with a prevalence of 3.23 per 100,000 people and a 5-year survival rate of only 7.2% in the United States [1]. Even with current multimodal treatment that involves maximal safe surgery, radiotherapy, and chemotherapy, the median survival is still less than 2 years [2,3]. Prior histopathological classification alone is no longer valid because some specimens do not show typical features, and various features can coexist within one tumour. Further, the method of categorisation is not precisely correlated with tumour behaviour. In the 2016 World Health Organization (WHO) classification, molecular profiles are integrated into brain tumour diagnoses owing to their prognostic significance [4]. Approximately 91% of all GBMs classified as isocitrate dehydrogenase (IDH) wild-type GBMs have a median survival of 1.2 years. In contrast, the median survival of 9% of IDH mutant GBMs is 3.6 years [5]. Given the different biological characteristics of IDH mutations, it has been suggested that GBM should be separated from both wild-type IDH and H3.3 G34 wild-type diffuse astrocytic glioma [6,7]. Moreover, rapid epigenetic and genetic discoveries have advanced our understanding of the molecular pathogenesis of GBMs. The Cancer Genome Atlas (TCGA) has summarised genomic expression data and classifies GBMs into three distinct transcriptional subtypes: classical, mesenchymal, and proneural. However, as these subtypes can vary within the same tumour, change over time, and differ in their microenvironments, their clinical significance remains unclear [8,9]. Both high inter-tumour and intra-tumour heterogeneity contribute to the therapeutic failure of GBMs. As such, extensive efforts have been made to identify predictive biomarkers that could also be new feasible therapeutic targets to improve the outcomes of GBMs [10]. 

Thioredoxin domain-containing protein 11 (TXNDC11), also known as EF-hand-binding protein 1 (EFP1), was first identified in 2005 as an interaction partner of the hydrogen peroxide-generating enzyme dual oxidase 1 in the thyroid [11]. This study aimed to establish that TXNDC11 is an oncogene in GBMs.

## 2. Results

### 2.1. High TXNDC11 Expression Is Associated with Worse Prognosis

Among the 86 patients who underwent glioma underwent surgery at Kaohsiung Medical University Chung-Ho Memorial Hospital, 64 and 22 were aged ≤60 years and >60 years, respectively. Twenty patients had low-grade gliomas (WHO grade II), while 66 had high-grade gliomas (WHO grades III-IV). The distribution of patients with high and low TXNDC11 expression according to the IHC staining results is shown in Figure 1A. There were no significant differences in age (*p* = 1), sex (*p* = 1), tumour size (*p* = 0.789), or Karnofsky Performance Score (KPS) (*p* = 0.771) between the patients with high and low TXNDC11 expression. However, patients with high TXNDC11 expression had significantly shorter survival (*p* < 0.001) (Figure 1B). In addition, these patients showed a predominance of high-grade gliomas (*p* < 0.001) (Table 1). Multivariate analysis showed that both the WHO grade and TXNDC11 expression were independent prognostic factors (Table 2).

### 2.2. TXNDC11 Knockdown Inhibits GBM Proliferation and Increases Temozolomide (TMZ) Sensitivity 

Real-time PCR showed that the mRNA levels of TXNDC11 in GBM cells such as GBM8401, U87, U87 IDH-1 mt, G5T, DBTRG-05MG, M059K, and A172 cells were generally higher than those in normal glial cells (Figure 2). Due to the notably elevated expression levels of TXNDC11 observed in GBM cells, particularly in GBM8401 and U87 cells, these two cell lines were selected for further investigation in subsequent experiments. siRNA knockdown of TXNDC11 resulted in a significantly decreased proliferation of GBM cells in GBM8401 (Figure 3A) and U87 cells (Figure 3B). However, knock-in TXNDC11 plasmid resulted in a significantly increased proliferation of GBM cells in GBM8401 (Figure 3C) and U87 cells (Figure 3D). TMZ effectively inhibited the growth of GBM cells. Knockdown of TXNDC11 expression resulted in decreased viability of GBM8401cells (Figure 4A) and U87 cells (Figure 4B). On the contrary, knock-in TXNDC11 plasmid could potentially enhance resistance to TMZ in GBM8401cells (Figure 4C) and U87 cells (Figure 4D).

### 2.3. TXNDC11 Facilitates Migration and Invasion and Downregulates Apoptosis of GBM Cells

In light of exploring the diverse impacts of TXNDC11, it is important to note that our study aimed to comprehensively understand its role in glioma cell behaviours. For this purpose, the wound healing assay was used to detect migration ability. In GBM8401 cells, both si-TXNDC11 #1 and #2 markedly inhibited the migratory capability at 16, 20, and 24 h (Figure 5A). In U87 cells, both si-TXNDC11 #1 and #2 markedly inhibited the migratory capability at 12 and 24 h (Figure 5B). However, over-expression of TXNDC11 through transfection with TXNDC11 plasmid significantly enhanced the migratory capability GBM8401 (Figure 5C) and U87 (Figure 5D). These results of the wound healing assays indicated that the protein expression of TXNDC11 affects the migration capability in GBM cells. On the other hand, the Transwell assay was used to detect invasion ability. Transwell assays revealed a significant attenuation in the invasive capacity of GBM8401 (Figure 6A) and U87 (Figure 6B) cells following the knockdown of TXNDC11. Knock-in TXNDC11 plasmid resulted in a significantly increased invasion ability in GBM8401 (Figure 6C) and U87 cells (Figure 6D). In addition, Annexin V staining was used to detect the percentage of apoptosis following flow cytometry. The results of Annexin V staining showed that the percentage of apoptosis was individually 0.32 ± 0.09, 14.39 ± 1.27, and 14.31 ± 0.92 in negative control, si-TXNDC11#1, and si-TXNDC11#2 group in GBM8401 cells (Figure 7A). In U87 cells, the percentage of apoptosis was individually 0.19 ± 0.02, 19.06 ± 2.30, and 22.88 ± 1.47 in negative control, si-TXNDC11#1, and si-TXNDC11#2 group (Figure 7B). The result supported the finding that depletion of TXNDC11 resulted in increased apoptosis of GBM cells. 

### 2.4. TXNDC11 Promotes Epithelial-Mesenchymal Transition and Affects Cell Cycle of GBM Cells

Western blotting of epithelial–mesenchymal transition (EMT)-related markers under different TXNDC11 expression levels showed that TXNDC11 knockdown in GBM8401 and U87 cells upregulated the expression of E-cadherin but downregulated the expression of N-cadherin (Figure 8). Therefore, we speculated that TXNDC11 may induce EMT in GBM cells. In addition, cyclin D1 expression was decreased under siRNA inhibition of TXNDC11 (Figure 8). Notably, the overexpression of TXNDC11 through the knock-in plasmid led to a distinct shift in EMT-related markers in GBM8401 and U87 cells. Specifically, the levels of E-cadherin, an epithelial marker, were downregulated, while N-cadherin, a mesenchymal marker, experienced upregulation (Figure 8). Furthermore, the impact of the knock-in TXNDC11 plasmid extended beyond EMT-related markers. A noteworthy outcome was the observed increase in cyclin D1 expression (Figure 8), signifying a potential acceleration of cell cycle progression and proliferation. The introduction of a knock-in TXNDC11 plasmid can distinctly reverse these phenomena, and this study sheds light on the intricate interplay between TXNDC11 and critical cellular processes.

### 2.5. TXNDC11 Knockdown Attenuated the Growth of GBM Cells In Vivo

The animal model used to examine the in vivo functions of TXNDC11 is shown in Figure 9A. The fluorescence intensity of mice injected with knockdown TXNDC11 cells was significantly lower than that of mice in the control group on days 14 (*p* < 0.01) and 21 (*p* < 0.001) (Figure 9B). The survival time of the TXNDC11 knockdown group was significantly longer than that of the control group (25.50 + 0.812 days vs. 20.75 + 0.978 days, *p* = 0.006) (Figure 9C). This indicates that TXNDC11 plays an important role in the regulation of tumour growth in vivo.

## 3. Discussion

Glioblastoma accounts for 14.5% of all CNS tumours and 57.7% of all gliomas and is the most common primary brain malignancy in adults. The incidence of GBM increases with age, with the highest rate among those aged 75–84 years. Older age, female sex, white race, and non-Hispanic ethnicity are associated with poor survival [1]. The initial treatment is a maximally safe resection. Although more advanced surgery helps prolong survival [12], it is challenging to take function-preserving and ill-defined tumour margins into account. Even with multimodal treatment, the median overall survival of GBMs ranges from 14.6 months to 16.7 months, and the progression-free survival ranges from 6.2 months to 7.5 months [13]. TMZ is one of the most important chemotherapeutic agents used to treat high-grade gliomas. However, its therapeutic effect is affected by DNA repair systems, especially the expression of O^6^-methylguanine-DNA methyltransferase. Accordingly, almost half of the patients have a poor response to TMZ treatment, and some ultimately develop drug resistance [14,15,16]. Almost all GBM patients experience recurrence. Unfortunately, no effective second-line treatment to prolong overall survival has been developed to date [17,18]. There has been extensive effort to develop targeted therapies and immunotherapies, but most studies are still in Phase I or II, and the small minority of phase III trials have not made a breakthrough [19,20]. Therefore, it is essential to understand the pathogenesis of GBMs and identify the potential oncogenes. 

The TXNDC family, a regulator of the redox status at distinct cellular locations, includes 17 members [21]. Several studies have investigated their role in cancer progression. For example, the expressions of TXNDC2, TXNDC3, and TXNDC6 have a significance in both testicular and systemic diffuse large B-cell lymphoma [22]. TXNDC5 is involved in protein folding and chaperone activity and is abnormally expressed in many cancers, such as non-small cell lung cancer, prostate cancer, gastric adenocarcinoma, colon cancer, and hepatocellular carcinoma [23]. Meanwhile, TXNDC9 promotes the progression of hepatocellular carcinoma and prostate cancer [24,25] and regulates apoptosis and autophagy in glioma and colorectal cancer [26,27]. TXNDC12 enhances EMT in liver cancer [28]. TXNDC17 is involved in chemotherapy resistance of ovarian cancer [29]. 

TXNDC11 can act as a disulfide reductase involved in endoplasmic reticulum-associated degradation [30]. Moreover, TXNDC11 is upregulated during endoplasmic reticulum stress and may trigger Par-4-mediated apoptosis [31]. The current study evaluated glioma patients treated in our institution between 2010 and 2020 and found that high TXNDC11 expression was associated with advanced-grade glioma. This result supported the assertion that TXNDC11 is potentially associated with malignancy. TXNDC11 is an intracellular redox protein that is linked to cellular stress and protein folding processes. Overexpression of TXNDC11 in certain contexts might lead to increased oxidative stress within cells, potentially triggering malignant behaviours such as proliferation, invasion, and metastasis. In fact, low TXNDC11 expression was associated with longer survival. Multivariate analysis revealed that TXNDC11 expression was an independent prognostic factor for gliomas. Consistent findings were observed in the study from Peng, P et al. [32]. In their study, TXNDC11 expression was association with age, gender, WHO grade, histological type, IDH1 mutation, 1p19q-codeletion-status, and overall survival time. Our in vitro study also showed that siRNA knockdown of TXNDC11 resulted in a marked inhibition of GBM cell growth and a more sensitive response to TMZ therapy. Furthermore, the invasion and migration of GBM cells were reduced when TXNDC11 expression was suppressed. On the contrary, TXNDC11 overexpression has been associated with potentially opposite effects, contributing to increased oxidative stress within cells, which could potentially trigger malignant behaviours such as proliferation, invasion, and metastasis.

EMT occurs during embryogenesis, tissue regeneration, and wound healing and is involved in tumour metastatic expansion, cancer stem cell differentiation, and treatment resistance [33,34]. E-cadherin often acts as a tumour suppressor. The loss of its expression promotes EMT and facilitates invasion and metastasis [35]. In this study, TXNDC11 knockdown led to increased E-cadherin expression and reduced N-cadherin expression. This supports the conclusion that TXNDC11 induces EMT in GBM cells. Annexin V staining and flow cytometry showed that apoptosis was increased when TXNDC11 expression was downregulated. Additionally, TXNDC11 knockdown attenuated the expression of cyclin D1, an essential regulator of the G1 to S phase transition in the cell cycle. In many cancers, the overactivation of CCDN1 drives cell proliferation [36]. These data support the role of TXNDC11 in cell cycle regulation in GBMs. Unfortunately, the sample size, consisting of 86 patients, could potentially introduce statistical biases and limitations to the generalizability of the findings. Additionally, one significant limitation pertains to the absence of clinical data documenting whether surgeries were complete resections or not, along with the lack of postoperative CT imaging confirmation of any residual tumour presence. This discrepancy between cellular results and clinical outcomes might result from these missing details.

In summary, high TXNDC11 expression is associated with high-grade glioma and poor prognosis. Further, TXNDC11 expression is an independent prognostic factor for gliomas. TXNDC11 knockdown inhibits the proliferation, invasion, migration and EMT of GBM cells and promotes apoptosis. Furthermore, a significant dimension emerges from the study, revealing that overexpression of TXNDC11 can indeed reverse these observed effects. These results suggest that TXNDC11 is a potential oncogene in GBMs and may be an emerging therapeutic target.

## 4. Materials and Methods

### 4.1. Sample and Preparation

This study was approved by the Institutional Review Board of Kaohsiung Medical University Hospital (KMUH-IRB-20210186) and was conducted according to the tenets of the Declaration of Helsinki. 

A total of 146 patients who underwent surgical treatment for GBM at the Kaohsiung Medical University Chung-Ho Memorial Hospital between 2010 and 2020 were evaluated. Among them, 60 patients with incomplete medical records or low-quality pathological results were excluded. Finally, 86 patients were evaluated. 

### 4.2. Tumour Immunohistochemistry

Tissues collected from each patient were fixed in formalin, embedded in paraffin, and cut into 3-μm sections. To retrieve antigens for immunohistochemical (IHC) staining, the samples were de-paraffinised, rehydrated, and autoclaved at 121 °C for 5 min in pH 6.0 citrate buffer. Subsequently, the sections were incubated with 3% hydrogen peroxide at room temperature for 10 min to block endogenous peroxidase activity. After washing with Tris buffer solution (TBS), the sections were incubated with the primary antibodies at 4 °C. The specimens were then washed with TBS and incubated with secondary antibodies for 30 min at room temperature. Finally, the specimens were incubated with 3,3-diaminobenzidine for 5 min, followed by Mayer’s haematoxylin counterstaining for 1 min. Immunohistochemically stained sections were evaluated based on the intensity of staining and the proportion of positively stained tumour tissue. The staining intensity was graded as 0 (zero, no staining), 1 (weak staining), 2 (moderate staining), or 3 (strong staining). If the extent of the stained tumour was zero, the section was scored as 0, whereas <10%, 10–50%, and >50% corresponded to 1, 2, and 3, respectively. The final index was generated by multiplying these two independent parameters (ranging from 0 to 9), and the cutoff value was 4. A final index of ≥4 indicated high expression, whereas an index of <4 indicated low expression [37]. 

### 4.3. Cell Lines and Cell Culture

All the cells were incubated in 5% CO_2_ at 37 °C. SVGp12, U87MG, GBM8401, GBM8901, DBTRG-05MG, G5T/VGH, and M059K were obtained from the Bioresource Collection and Research Center (BCRC), whereas A172 was obtained from the American Type Culture Collection. GBM8401, GBM8901, and DBTRG-05MG cells were cultured in 90% Roswell Park Memorial Institute medium supplemented with 10% foetal bovine serum (FBS). SVGp12 and U87MG cell lines were cultured in minimum essential medium containing 10% FBS. The G5T and A172 cell lines were cultured in Dulbecco’s Modified Eagle Medium (DMEM) supplemented with 10% FBS. M059K cells were grown in DMEM-F12 supplemented with 10% FBS. The SVGp12 cell line was isolated from normal glial cells and was used as a normal control.

### 4.4. Real-Time PCR

Total RNA was extracted using the PureLink^TM^ RNA Mini Kit, and its concentration was quantified using the Quant-iT^TM^ RiboGreen^TM^ RNA Assay Kit. Real-time polymerase chain reaction (PCR) was performed as denaturation at 95 °C for 3 min and then 50 cycles of 95 °C for 5 s and 60 °C for 30 s. The PCR primer sequences were TXNDC11 forward primer: 5′-CAAGC AACGT TGTTT AACTA-3′ and reverse primer: 5′-CGTAA CGAAT AGTTA AACAAC-3′, and GAPDH forward primer: 5′-GGT CAC CAG GGC TGC TTT TA-3′ and reverse primers: 5′-GGA TCT CGC TCC TGG AAG ATG-3′.

### 4.5. Cell Proliferation Assay

The cells were seeded into a 24-well plate at 30,000 cells per well and incubated in 5% CO_2_ at 37 °C for 24 h. After 24, 48, and 72 h of co-culture with or without small interferon RNA (siRNA), cell populations were counted using the 3-(4,5-dimethylthiazol-2-yl)-2,5-diphenyltetrazolium bromide assay.

### 4.6. Migration Assay In Vitro

Cell migration was assessed using a wound-healing assay (ibidi; 80209). Each well contained 70 μL cell culture medium (5 × 10^5^ cells/mL) and was cultured at 37 °C with or without siRNA nonsense siRNA. The healing conditions were observed under an optical microscope at 0, 16, 20, and 24 h.

### 4.7. Cell Invasion Assay In Vitro

The cell invasion assay was performed using a Transwell (CORNING; COR3452) invasion assay kit. There were 5 × 10^5^ cells per insert. The lower chamber was filled with 2 mL of medium with or without siRNA or nonsense siRNA. Cells that remained on the upper surface were removed after 24 h of incubation. Those invading the Transwell to the bottom of the insert were fixed with methanol, stained with crystal violet, and counted in six selected high-power fields under a microscope. 

### 4.8. Transfection

For siRNA transfection, 1000 cells were seeded onto a 6-well plate and incubated at 37 °C overnight. We then transfected 5 μm siRNA with DharmaFECT^TM^ transfection reagent following the manufacturer’s protocol. The sequence for TXNDC11 siRNA#1 was GCAUAGAAUGCAGCAAUUU[dT][dT], and the sequence for #2 was GCAUGUUGCAGGACCAUAA[dT][dT]. After transfection with siRNA, the cells were cultured for 3 days before use.

### 4.9. Western Blotting

After lysis with lysis buffer, 20 μg of protein per sample was loaded onto a lane of a sodium dodecyl sulphate-polyacrylamide gel for electrophoresis. The proteins were electrotransferred to polyvinylidene fluoride membranes. The transferred membrane was treated with blocking buffer and incubated with primary antibodies (E-cadherin [20874-1-AP; Proteintech, Chicago, IL, USA)], N-cadherin [22018-1-AP; Proteintech, Chicago, IL, USA], Cyclin D1 [60186-1-lg; Proteintech, Chicago, IL, USA], β actin [MAB1501R; Millipore, Burlington, VT, USA]) for 2 h at room temperature. Secondary antibodies (goat anti-rabbit [AP132P; Millipore, Burlington, VT, USA] and goat anti-mouse [AP124P; Millipore, Burlington, VT, USA]) were then added, and the cells were incubated for 90 min at room temperature. An enhanced chemiluminescence solution (205–14,621; Revvity, Burlington, VT, USA) was used to detect specific bands using a MINICHEMI (Thermo) system.

### 4.10. Apoptosis Analysis

By using a Muse^®^ Cell Cycle Kit, GBM8401 and U87 cells were fixed with cold 70% ethanol for 72 h at −20 °C. Then, the Muse^®^ Cell Cycle reagent was added, and the results were analysed using a Guava^®^ Muse^®^ Cell Analyser. For apoptosis analysis, 100 μL Muse ^®^ Annexin V and Dead Cell reagent was mixed with 100 μL (10^5^ cells/mL) of cells at room temperature, and the results were analysed 20 min later. 

### 4.11. Animal Model

All animal experiments were approved by the Committee of Institutional Animal Research of Kaohsiung Medical University (IACUC 111011). All applicable international, national, and/or institutional guidelines for the care and use of animals were followed.

GBM8401 cells including fluorescent (1 × 10^5^ cells/10 μL) were implanted intracranially in the striatum of immunodeficient mice from LASCO Laboratory Animal Center (Taipei, Taiwan). All mice were housed under a constant temperature (24 °C) and regular light/dark cycles (12  h/12  h), with free access to a standard diet. The control group was injected with GBM8401 cells (n = 12), and the knockdown TXNDC11 group was injected with knockdown TXNDC11 GBM8401 cells (n = 12). Mice were anaesthetised with isoflurane, and fluorescence was detected using the Xenogen IVISR Spectrum Noninvasive Quantitative Molecular Imaging System (J&H; IVIS Lumina LT 2D) at 7, 14, and 21 days after injection with GBM cells. 

### 4.12. Statistical Analysis

TXNDC11 expression in GBM cells was analysed by IHC staining and assessed using the Chi-square test. Overall survival was evaluated using the Kaplan–Meier method. Univariate and multivariate Cox regression models were used to understand the relationships between the different variables. All statistical analyses were performed using SPSS version 24.0. *p* < 0.05 was considered statistically significant.

## Figures and Tables

**Figure 1 ijms-24-13367-f001:**
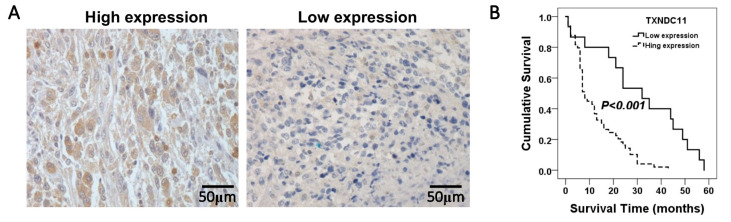
Immunohistochemistry staining for TXNDC11 expression in glioma. Representative immunohistochemical staining results for gliomas with (**A**) high and low TXNDC11 expression. (**B**) Analysis of TXNDC11 expression using the Kaplan–Meier method with the log-rank test.

**Figure 2 ijms-24-13367-f002:**
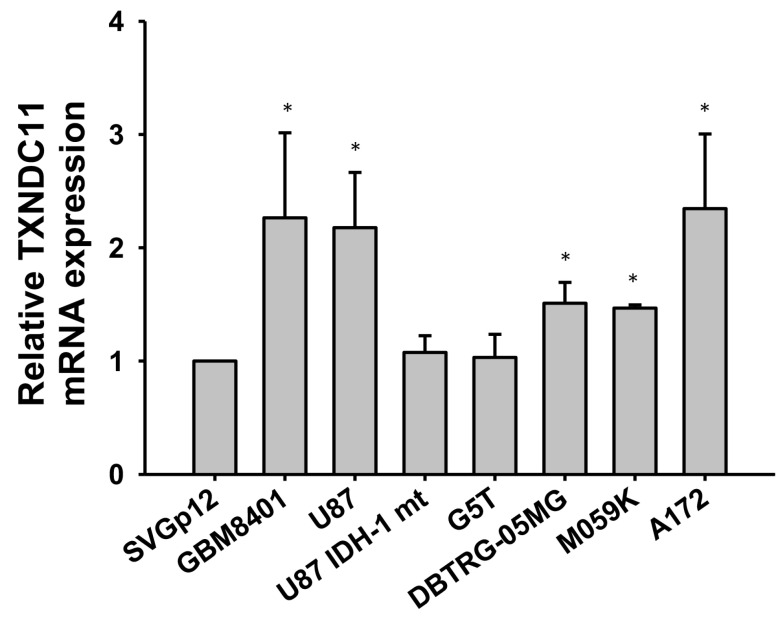
Comparison of mRNA TXNDC11 expression between normal glial cells and GBM cell lines. Comparison of TXDNC11 mRNA expression in normal glial cells (SVGp12) and GBM cells (GBM8401, U87, U87 IDH-1 mt, G5T, DBTRG-05MG, M059K, and A172 cells) using real-time PCR. * *p* < 0.05 compared to SVGp12.

**Figure 3 ijms-24-13367-f003:**
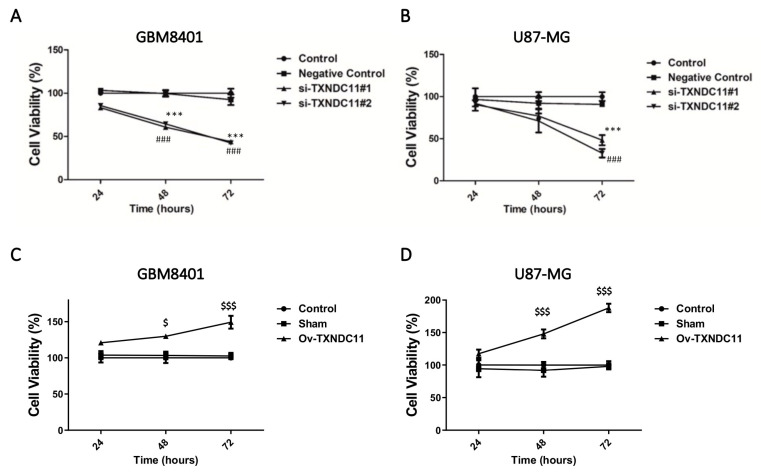
Effect of TXNDC11 on viability of GBM cells. (**A**) After transfection with TXNDC11 siRNA, cell viability is detected by MTT assay in GBM8401 cells at 24, 48, and 72 h. (**B**) After transfection with TXNDC11 siRNA, cell viability is detected by MTT assay in U87-MG cells at 24, 48, and 72 h. (**C**) After transfection with TXNDC11 plasmid, cell viability is detected by MTT assay in GBM8401 cells at 24, 48, and 72 h. (**D**) After transfection with TXNDC11 plasmid, cell viability is detected by MTT assay in U87-MG cells at 24, 48, and 72 h. *** *p* < 0.001 compared between control group and si-TXNDC11#1 group. ### *p* < 0.001 compared between control group and si-TXNDC11#2 group. $ *p* < 0.05 and $$$ *p* < 0.001 compared between control group and Ov-TXNDC11 group.

**Figure 4 ijms-24-13367-f004:**
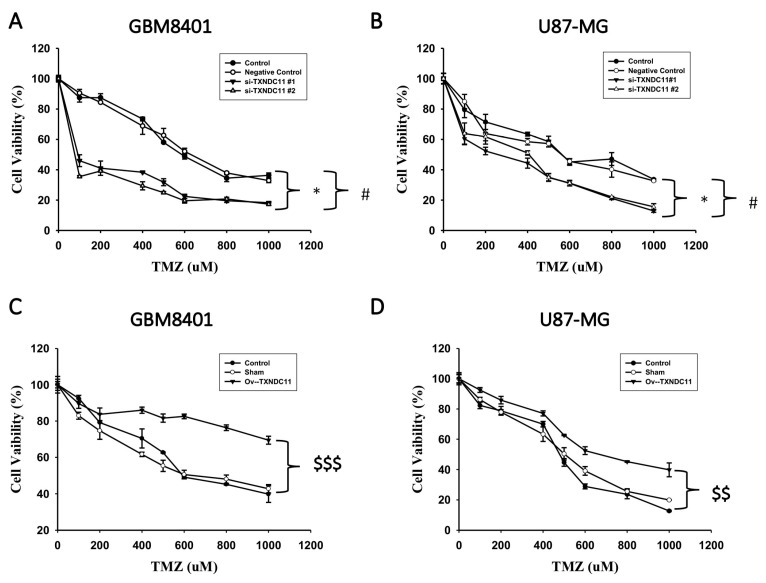
Effect of combination of siRNA knock-down or plasmid knock-in of TXNDC11 and TMZ treatment on viability of GBM cells. (**A**) After siRNA transfection, GBM8401 cells are seeded in 24 wells and treated with different doses of TMZ. At 72 h, cell viability is determined using the MTT assay. (**B**) After siRNA transfection, U87-MG cells are seeded in 24 wells and treated with different doses of TMZ. At 72 h, cell viability is determined using the MTT assay. (**C**) After plasmid transfection, GBM8401 cells are seeded in 24 wells and treated with different doses of TMZ. At 72 h, cell viability is determined using the MTT assay. (**D**) After plasmid transfection, U87-MG cells are seeded in 24 wells and treated with different doses of TMZ. At 72 h, cell viability is determined using the MTT assay. * *p* < 0.05 compared between control group and si-TXNDC11#1 group. # *p* < 0.001 compared between control group and si-TXNDC11#2 group. $$ *p* < 0.01 and $$$ *p* < 0.001 compared between control group and Ov-TXNDC11 group.

**Figure 5 ijms-24-13367-f005:**
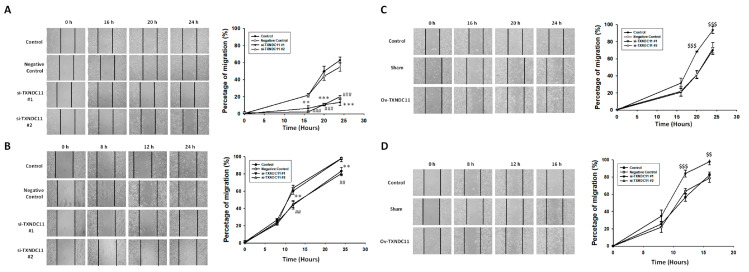
Wound healing analysis for the effect of siRNA knock-down or plasmid knock-in of TXNDC11 on migration ability of GBM cells at 0, 8, 12, and 24 h. (**A**,**B**) Wound healing analysis of GBM8401 and U87MG cells transfected with TXNDC11 siRNA. (**C**,**D**) Wound healing analysis of GBM8401 and U87MG cells transfected with TXNDC11 plasmid. ** *p* < 0.01 and *** *p* < 0.001 compared between control group and si-TXNDC11#1 group. ## *p* < 0.01 and ### *p* < 0.001 compared between control group and si-TXNDC11#2 group. $$ *p* < 0.01 and $$$ *p* < 0.001 compared between control group and Ov-TXNDC11 group.

**Figure 6 ijms-24-13367-f006:**
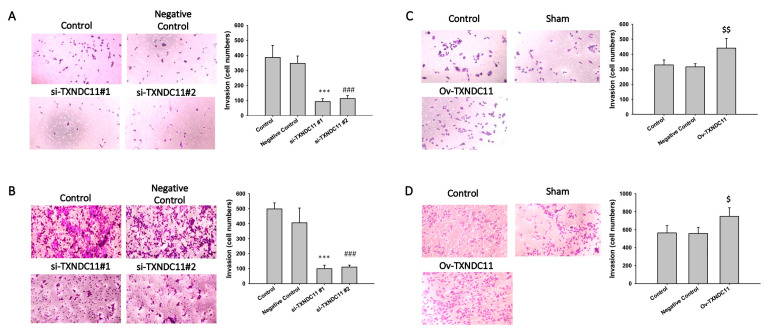
Effect of siRNA knockdown of TXNDC11 on invasion ability of GBM cells at 24 h. (**A**,**B**) Transwell invasion analysis of GBM8401 and U87MG cells transfected with TXNDC11 siRNA. (**C**,**D**) Transwell invasion analysis of GBM8401 and U87MG cells transfected with TXNDC11 plasmid. (100×) *** *p* < 0.001 compared between control group and si-TXNDC11#1 group. ### *p* < 0.001 compared between control group and si-TXNDC11#2 group. $ *p* < 0.05 and $$ *p* < 0.01 compared between control group and Ov-TXNDC11 group.

**Figure 7 ijms-24-13367-f007:**
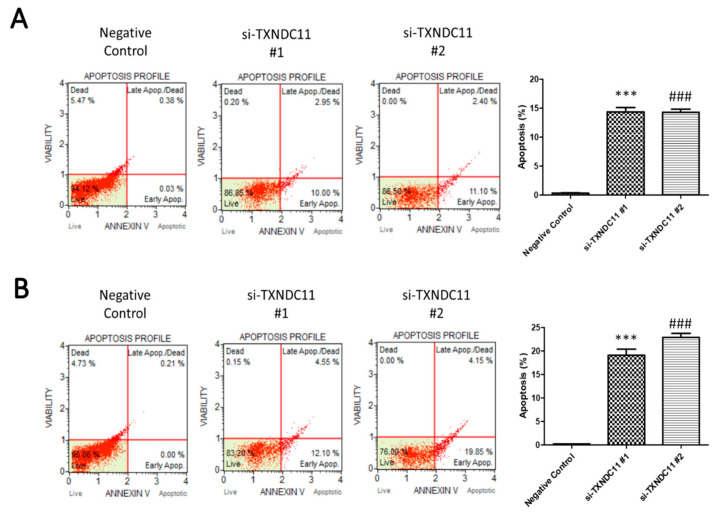
Effect of siRNA knockdown of TXNDC11 on the apoptosis of GBM cells. Cell apoptosis analysis of (**A**) GBM8401 and (**B**) U87MG cells transfected with TXNDC11 siRNA following flow cytometry. *** *p* < 0.001 compared between negative control group and si-TXNDC11 #1. ### *p* < 0.001 compared between negative control group and si-TXNDC11 #2.

**Figure 8 ijms-24-13367-f008:**
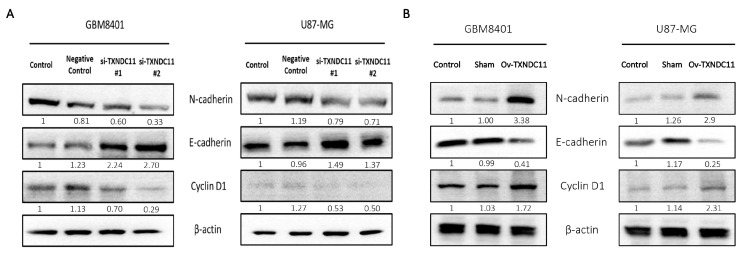
Mechanism of TXNDC11 in GBM cells. Western blot analysis of the protein expression of E-cadherin, N-cadherin, and cyclin D1 with (**A**) TXNDC11 siRNA or (**B**) TXNDC11 plasmid in GBM8401 cells and U87MG cells.

**Figure 9 ijms-24-13367-f009:**
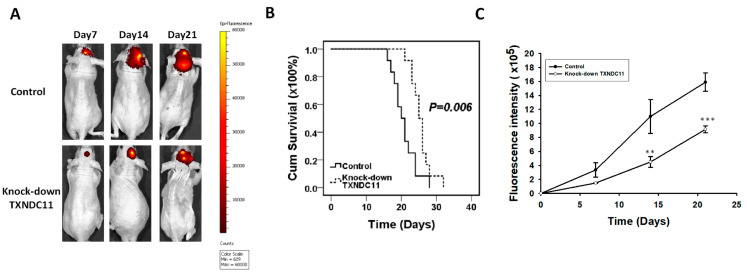
Effects of TXNDC11 on progression of GBM cells in vivo. Fluorescence is measured following the intracranial implantation of GBM8401 cells. (**A**) IVIS images, (**B**) intensity, and (**C**) survival times were compared between TXNDC11 knockdown and control groups. ** *p* < 0.01 and *** *p* < 0.001 compared between control group.

**Table 1 ijms-24-13367-t001:** Expression of TXNDC11 correlated with clinicopathological parameters in gliomas.

		TXNDC11 Expression		*p* Value
		Low	High	
Age				1
<60	64 (74.4%)	14 (16.3%)	50 (58.1%)	
≥60	22 (25.6%)	4 (4.7%)	18 (20.9%)	
Gender				1
Male	48 (55.8%)	10 (11.6%)	38 (44.2%)	
Female	38 (44.2%)	8 (9.3%)	30 (34.9%)	
WHO grade				<0.001
II	20 (23.3%)	12 (14.0%)	8 (9.3%)	
III/IV	66 (76.7%)	6 (7.0%)	60 (69.8%)	
Tumour Size				0.789
<3 cm	54 (62.8%)	12 (14.0%)	42 (48.8%)	
≥3 cm	32 (37.2%)	6 (7.0%)	26 (30.2%)	
KPS				0.771
<70	61 (70.9%)	12 (14.0%)	49 (57.0%)	
≥70	25 (29.1%)	6 (7.0%)	19 (22.1%)	

**Table 2 ijms-24-13367-t002:** Univariate and multivariate Cox regression analyses of prognostic indicators in patients with gliomas.

	Univariate		Multivariate	
	HR (95% CI)	*p*	HR (95% CI)	*p*
Age	0.755 (0.419–1.360)	0.349		
Gender	0.880 (0.534–1.451)	0.616		
WHO grade	0.316 (0.160–0.624)	0.01	0.452 (0.220–0.930)	0.031
Tumour size	1.295 (0.757–2.217)	0.345		
Radiotherapy	1.183 (0.714–1.959)	0.514		
TMZ	0.987 (0.599–1.626)	0.958		
KPS	1.387 (0.798–2.408)	0.236		
TXNDC11	0.248 (0.117–0.524)	<0.001	0.334 (0.153–0.729)	0.006

## Data Availability

No new data were created or analysed in this study. Data sharing is not applicable to this article.

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
