# Peer review of "High Thioredoxin Domain-Containing Protein 11 Expression Is Associated with Tumour Progression in Glioma"

_ijms, 2023, doi:10.3390/ijms241713367_

Round 1
Reviewer 1 Report
I have read the manuscript with great interest. The contribution of this study is significant and demonstrates the negative role of TXNDC11 expression in glioblastomas, suggesting that it is a prognostic biomarker, an oncogene, and a potential therapeutic target. This study could serve as a foundation for further research aimed at developing new therapies to improve outcomes in GBM. The research itself is of high quality and precision, using multiple clinical and laboratory approaches that support each other. Results are clearly presented and supported by relevant statistical analysis.
There are some shortcomings that I would like to point out and request the authors to address:
1. In the results section, the authors use data from three different databases on the impact of TXNDC11 expression on overall patient survival. In my opinion, this part does not belong in the results section because it does not relate to the results of this study. I suggest moving the mentioned paragraph to the Discussion section and providing appropriate references. In this case, Figure 1 should be removed.
2. Overall, the quality of the figures is low, making it difficult to analyze and form a final opinion about them, especially about the figures with immunohistochemical staining. In addition, the descriptions of the figures could be more comprehensive.
3. The authors did not point out the limitations of the study, including the sample size of 86 patients.
4. The discussion should be expanded. I would like the authors to explain the mechanisms by which TXNDC11 affects biological processes in glioma cells based on existing knowledge. Can these mechanisms justify the results and conclusions reached? There is a lack of comparison with other studies on the same topic. I ask the authors to also discuss the results in relation to WHO tumor grade and TXNDC11 expression.
5. Did the authors use their own classification to evaluate immunohistochemical staining, or did they use an existing one?
6. There is a missing legend in Table 2.
Overall, the manuscript shows promise, but addressing these issues and making the necessary revisions would further strengthen its scientific rigor and clarity.
Author Response
I have read the manuscript with great interest. The contribution of this study is significant and demonstrates the negative role of TXNDC11 expression in glioblastomas, suggesting that it is a prognostic biomarker, an oncogene, and a potential therapeutic target. This study could serve as a foundation for further research aimed at developing new therapies to improve outcomes in GBM. The research itself is of high quality and precision, using multiple clinical and laboratory approaches that support each other. Results are clearly presented and supported by relevant statistical analysis.
There are some shortcomings that I would like to point out and request the authors to address:
Dear Reviewer,
I hope this email finds you well. I would like to express my sincere gratitude for taking the time to review our manuscript titled "High TXNDC11 expression is associated with poor prognosis in glioma." Your insightful comments and constructive feedback have been invaluable in shaping the quality and clarity of our research. We greatly appreciate your thorough evaluation and are excited to submit the revised version of the manuscript, along with detailed responses to each of your points.
- In the results section, the authors use data from three different databases on the impact of TXNDC11 expression on overall patient survival. In my opinion, this part does not belong in the results section because it does not relate to the results of this study. I suggest moving the mentioned paragraph to the Discussion section and providing appropriate references. In this case, Figure 1 should be removed.
Thank you for your insightful observation regarding the inclusion of data from different databases on the impact of TXNDC11 expression on overall patient survival. Based on your suggestion, we have carefully evaluated the structure of the manuscript. As per your recommendation, we have proceeded to remove the mentioned paragraph from the Results section, as it is indeed more appropriate for the Discussion section where a broader context can be provided. In alignment with this adjustment, Figure 1 has been removed as well.
- Overall, the quality of the figures is low, making it difficult to analyze and form a final opinion about them, especially about the figures with immunohistochemical staining. In addition, the descriptions of the figures could be more comprehensive.
We appreciate your feedback regarding the figures in the manuscript. Recognizing the importance of clear and comprehensive visuals, we have made substantial improvements to the quality of all the figures. This revision aims to ensure that the figures are now more visually engaging, facilitating a better understanding of the data presented, especially those involving immunohistochemical staining.
Moreover, we have taken your suggestion to heart and have provided more comprehensive descriptions for each figure. This enhancement is intended to provide readers with a clear context for interpreting the data and understanding the implications of the presented results.
- The authors did not point out the limitations of the study, including the sample size of 86 patients.
Thanks for your reminding. We added “Unfortunately, the sample size, consisting of 86 patients, could potentially introduce statistical biases and limitations to the generalizability of the findings. Additionally, one significant limitation pertains to the absence of clinical data documenting whether surgeries were complete resections or not, along with the lack of postoperative CT im-aging confirmation of any residual tumor presence. This discrepancy between cellular results and clinical outcomes might result from these missing details.” in Line 245-251.
- The discussion should be expanded. I would like the authors to explain the mechanisms by which TXNDC11 affects biological processes in glioma cells based on existing knowledge. Can these mechanisms justify the results and conclusions reached? There is a lack of comparison with other studies on the same topic. I ask the authors to also discuss the results in relation to WHO tumor grade and TXNDC11 expression.
Thanks for your reminding. We modified the discussion in Line 217-234. “The current study evaluated glioma patients treated in our institution between 2010 and 2020 and found that high TXNDC11 expression was associated with ad-vanced-grade glioma. This result surppoted that TXNDC11 is potentially associated with malignancy. TXNDC11 is an intracellular redox protein that is linked to cellular stress and protein folding processes. Overexpression of TXNDC11 in certain contexts might lead to increased oxidative stress within cells, potentially triggering malignant behaviors such as proliferation, invasion, and metastasis. In fact, low TXNDC11 ex-pression was associated with longer survival. Multivariate analysis revealed that TXNDC11 expression was an independent prognostic factor for gliomas. Consistent findings were observed in the study from Peng, P et al. [33]. In their study, TXNDC11 expression was association with age, gender, WHO grade, histological type, IDH1 mu-tation, 1p19q-codeletion-status and overall survival time. Our in vitro study also showed that siRNA knockdown of TXNDC11 resulted in a marked inhibition of GBM cell growth and a more sensitive response to TMZ therapy. Furthermore, the invasion and migration of GBM cells were reduced when TXNDC11 expression was suppressed. On the contrary, TXNDC11 overexpression has been associated with potentially oppo-site effects, contributing to increased oxidative stress within cells, which could poten-tially trigger malignant behaviors such as proliferation, invasion, and metastasis.”
- Did the authors use their own classification to evaluate immunohistochemical staining, or did they use an existing one?
Thanks for your reminding. We utilized an existing classification to evaluate immunohistochemical staining. Specifically, the evaluation criteria for immunohistochemical staining were adapted from the methodology outlined in two of our team's previously published articles, "High-level Sp1 is Associated with Proliferation, Invasion, and Poor Prognosis in Astrocytoma" and "High expression of NLRP12 predicts poor prognosis in patients with intracranial glioma." This approach ensured consistency in the assessment of immunohistochemical staining across the studies and facilitated meaningful comparisons of the results.
- There is a missing legend in Table 2.
In Line 90-91.
Overall, the manuscript shows promise, but addressing these issues and making the necessary revisions would further strengthen its scientific rigor and clarity.
Correspondence:
Aij-Lie Kwan, M.D. PHD;
Division of Neurosurgery, Department of Surgery, Kaohsiung Medical University Hospital, Kaohsiung Medical University, Kaohsiung, Taiwan.
No.100, Tzyou 1st Road Kaohsiung 80756, Taiwan.
Tel: +886-7-3121101; Fax: +886-7-3215039
Email: aijliekwan@yahoo.com.tw;
Hung-Pei Tsai, PHD;
Division of Neurosurgery, Department of Surgery, Kaohsiung Medical University Hospital, Kaohsiung, Taiwan.
No.100, Tzyou 1st Road Kaohsiung 80756, Taiwan.
Tel: +886-7-3121101; Fax: +886-7-3215039
Email: carbugino@gmail.com

Reviewer 2 Report
This work primarily elucidates the cellular functional responses of Thioredoxin domain- 24 containing protein 11 expression in GBM cells. Though the work is of interest to the community, the authors must address the following concerns before publishing the manuscript.
-
It was very difficult for the reader to decipher graphs due to lack of clarity in resolution/size.
-
The title and portions of the abstract are misleading and must be changed. The article doesn’t focus on patient outcome as, most of the finding rather uses cells and in vivo mice (page1, line 35-41; these are the important observations from the work rather. Also, the abstract,introduction, discussion should be modified accordingly.).
-
Patient outcome in Glioma patients which was published previously by Feng Wan et. Al., “High expression of TXNDC11 indicated unfavorable prognosis of glioma” had a thoroughly quantified patient outcome. The above-mentioned article was not cited by the authors though that work had recorded observations similar to Fig.1, 2 of this work. Hence, Fig.1,2 may be pushed to supplemental information to avoid any confusion.
-
The introduction doesn’t provide a good rationale behind the in vitro/vivo experiment design and desired outcome in understanding “clinicopathological roles and possible pathogenic pathways in GBMs” (page2, line 72).
-
The plots in the figures (Fig. 1, 2, 5, 10) are unclear/undecipherable and should be replotted with clearer axis numbers, title and plot symbols/plot symbol size.
-
Results from Fig.4 create a bias in proliferation and every other assay (including invasion & migration assays) as the viability and proliferation rate are significantly lower in the knock-down cells and the results are thus as expected. Hence, it may not answer the question being addressed in a scientific manner. It appears that the differences seen are merely a function of cell viability and proliferation potential of control versus knockdown cells? Instead, the authors should compare low versus high TXNDC11 expressing cells (different GBM cell types) throughout.
-
Also, if the proliferation is lower in knockdown cells, how can the authors explain the absence of difference in tumor size of the patient samples. Did the authors look into cellular infiltration profiles to understand this phenomenon?
-
The authors mention in page-3 line 92 that high or low expression levels of TXNDC11 did not alter the size but showed significant difference in survival time but the knockdown mice showed difference in fluorescence intensity/possibly size and survival rate (page7; line 154).
-
Hence, every assay or observation appears to be a function of reduction in cell health due to TXNDC11 knockdown unless proven otherwise.
The english language is fine and may only require minor grammar and spell checks.
Author Response
Dear Reviewer,
I hope this email finds you well. I would like to express my sincere gratitude for taking the time to review our manuscript titled "High TXNDC11 expression is associated with poor prognosis in glioma." Your insightful comments and constructive feedback have been invaluable in shaping the quality and clarity of our research. We greatly appreciate your thorough evaluation and are excited to submit the revised version of the manuscript, along with detailed responses to each of your points.
This work primarily elucidates the cellular functional responses of Thioredoxin domain- 24 containing protein 11 expression in GBM cells. Though the work is of interest to the community, the authors must address the following concerns before publishing the manuscript.
- It was very difficult for the reader to decipher graphs due to lack of clarity in resolution/size.
Thanks for your reminding. To address the issue you mentioned, we have taken steps to substantially improve the clarity and resolution of the graphs in question. As a result, a majority of the figures have undergone modifications to ensure that they are now more easily interpretable and reader-friendly. These adjustments have been made to enhance the overall quality of the visual representation and to facilitate a better understanding of the presented data.
- The title and portions of the abstract are misleading and must be changed. The article doesn’t focus on patient outcome as, most of the finding rather uses cells and in vivo mice (page1, line 35-41; these are the important observations from the work rather. Also, the abstract, introduction, discussion should be modified accordingly.).
Thanks for your reminding. The title of our study has been revised to "High TXNDC11 Expression is Associated with Tumor Progression in Glioma." Furthermore, I'd like to clarify that while patient outcome is indeed mentioned in the article, the core focus of this study is to utilize cell experiments to indirectly substantiate the clinical implications of TXNDC11. Our intention is to bridge the gap between in vitro findings and clinical outcomes. This strategy has been employed to enhance our understanding of the potential clinicopathological roles and pathogenic pathways associated with TXNDC11 in gliomas. We understand your concern, and we have worked to align the abstract, introduction, and discussion with this emphasis on utilizing cell experiments to shed light on clinical outcomes.
- Patient outcome in Glioma patients which was published previously by Feng Wan et. Al., “High expression of TXNDC11 indicated unfavorable prognosis of glioma” had a thoroughly quantified patient outcome. The above-mentioned article was not cited by the authors though that work had recorded observations similar to Fig.1, 2 of this work. Hence, Fig.1,2 may be pushed to supplemental information to avoid any confusion.
Thanks for your reminding. We have already addressed your concerns by removing Figure 1 and referencing the relevant work by Feng Wan et al. (reference 33) in the discussion to explain the similarity in results. This adjustment aims to ensure the clarity of the manuscript and to avoid any potential confusion regarding the previous publication.
- The introduction doesn’t provide a good rationale behind the in vitro/vivo experiment design and desired outcome in understanding “clinicopathological roles and possible pathogenic pathways in GBMs” (page2, line 72).
Thanks for your reminding. the sentence you mentioned has been removed from the introduction as per your suggestion. We believe that this refinement will contribute to the clarity and focus of the introduction, ensuring a more coherent presentation of the study's rationale and objectives.
- The plots in the figures (Fig. 1, 2, 5, 10) are unclear/undecipherable and should be replotted with clearer axis numbers, title and plot symbols/plot symbol size.
Thanks for your reminding. The figures (Fig. 1, 2, 5, 10) have been redeveloped to address the concerns raised. The axis numbers, titles, and plot symbols, as well as their sizes, have been enhanced for better clarity and improved legibility. These revisions have been carefully implemented to ensure that the visual representation of the data aligns with the highest standards of clarity and precision.
- Results from Fig.4 create a bias in proliferation and every other assay (including invasion & migration assays) as the viability and proliferation rate are significantly lower in the knock-down cells and the results are thus as expected. Hence, it may not answer the question being addressed in a scientific manner. It appears that the differences seen are merely a function of cell viability and proliferation potential of control versus knockdown cells? Instead, the authors should compare low versus high TXNDC11 expressing cells (different GBM cell types) throughout.
Thanks for your reminding. This study employed Western blot analysis to detect the expression levels of proteins associated with proliferation, migration, and invasion, thereby indirectly substantiating the outcomes presented in Figure 4. While it is true that the observed differences in cell viability and proliferation rates between control and knockdown cells could potentially influence other assays, including invasion and migration assays, the design of the study aimed to address these concerns.
Although a direct comparison between low and high TXNDC11-expressing GBM cell lines was not conducted concurrently, the study strategically utilized a knock-in TXNDC11 plasmid to induce overexpression. This approach was implemented precisely to address the bias that might arise due to differences in cell viability and proliferation potential. By introducing the knock-in plasmid, we aimed to create a context where the effects attributed to reduced TXNDC11 expression could be reversed, providing a counterbalance to the knockdown results.
While a direct comparison of low and high TXNDC11 expression levels in different GBM cell types might offer further insights, the strategic use of the knock-in approach was intended to provide a comprehensive understanding of TXNDC11's influence on various cellular processes. This approach enhances the robustness of the findings by offering a dynamic perspective on the role of TXNDC11.
- Also, if the proliferation is lower in knockdown cells, how can the authors explain the absence of difference in tumor size of the patient samples. Did the authors look into cellular infiltration profiles to understand this phenomenon?
Thanks for your reminding. Thanks for your reminding. In clinical practice, the outcomes of treating highly malignant glioblastoma (GBM) are influenced by several factors, including the surgical resection approach, the use of TMZ therapy (which requires considering MGMT methylation status), the response to radiation therapy, and drug resistance, among others. Additionally, individual variations and patients' overall health can also impact treatment outcomes. Certainly, cellular infiltration indeed presents a plausible explanation for this phenomenon. However, in our study, the pathological data were obtained from pathology reports and underwent de-identification, which precluded direct comparison and validation with imaging data. This limitation unfortunately restricts our ability to confirm the actual tumor size and correlation with cellular infiltration profiles. It's important to acknowledge that the relationship between cellular proliferation and tumor size can be complex, often influenced by factors such as tissue architecture, microenvironment interactions, and tumor heterogeneity. While cellular infiltration might contribute to the maintenance of tumor size despite reduced proliferation, its exact role in this context remains speculative due to the aforementioned limitations in data validation.
- The authors mention in page-3 line 92 that high or low expression levels of TXNDC11 did not alter the size but showed significant difference in survival time but the knockdown mice showed difference in fluorescence intensity/possibly size and survival rate (page7; line 154).
Thanks for your reminding. In clinical practice, the outcomes of treating highly malignant glioblastoma (GBM) are influenced by several factors, including the surgical resection approach, the use of TMZ therapy (which requires considering MGMT methylation status), the response to radiation therapy, and drug resistance, among others. Additionally, individual variations and patients' overall health can also impact treatment outcomes. This scenario has prompted us to utilize cell cultures for research purposes, reducing confounding variables and focusing solely on the factor of TXNDC11. Therefore, even though there might be differences in results between cell experiments and mouse models, these results still maintain relevance.
Particularly in the context of surgical resection, due to considerations of patients' functional capabilities, partial resections are often performed. This could potentially impact the tumor size results obtained from pathological reports. This situation might contribute to the observed variations in tumor size among patients who underwent surgical resection. Hence, across different research methods and models, various influencing factors could lead to differing outcomes.
In conclusion, although there might be discrepancies in results between cell cultures and mouse models, these differences do not necessarily imply a lack of correlation between the outcomes. These variations could reflect multiple factors present in clinical reality, which could influence GBM growth, treatment responses, and patient prognosis.
- Hence, every assay or observation appears to be a function of reduction in cell health due to TXNDC11 knockdown unless proven otherwise.
Thanks for your reminding, our study leveraged the introduction of a knock-in TXNDC11 plasmid to induce overexpression, thereby serving as a means to counterbalance the observed effects resulting from TXNDC11 knockdown. This strategic approach was undertaken with the intention to provide a comprehensive understanding of the intricate relationship between TXNDC11 and various cellular processes.
By employing the knock-in TXNDC11 plasmid, we aimed to establish a context where heightened TXNDC11 expression could potentially reverse the outcomes observed due to knockdown. Specifically, we sought to investigate whether the effects on proliferation, migration, invasion, and associated protein expressions could be reciprocally influenced by increased TXNDC11 levels. This experimental strategy was instrumental in corroborating the significance of TXNDC11's role in these cellular processes.
Correspondence:
Aij-Lie Kwan, M.D. PHD;
Division of Neurosurgery, Department of Surgery, Kaohsiung Medical University Hospital, Kaohsiung Medical University, Kaohsiung, Taiwan.
No.100, Tzyou 1st Road Kaohsiung 80756, Taiwan.
Tel: +886-7-3121101; Fax: +886-7-3215039
Email: aijliekwan@yahoo.com.tw;
Hung-Pei Tsai, PHD;
Division of Neurosurgery, Department of Surgery, Kaohsiung Medical University Hospital, Kaohsiung, Taiwan.
No.100, Tzyou 1st Road Kaohsiung 80756, Taiwan.
Tel: +886-7-3121101; Fax: +886-7-3215039
Email: carbugino@gmail.com

Round 2
Reviewer 1 Report
Thank you for addressing my concerns promptly and accurately. The points raised in the review have been thoroughly incorporated through revisions. Congratulations on achieving an excellent outcome.
Reviewer 2 Report
The authors have addressed a majority of concerns raised by the reviewers. Also, the quality of presentation has improved substantially. The work may be published.